# Room- and High-Temperature Wear Resistance of MCrAlY Coatings Deposited by Detonation Gun (D-gun) and Supersonic Plasma Spraying (SSPS) Techniques

**Mehmet Kilic** [1,*], **Dervis Ozkan** [2] , **Mustafa Sabri Gok** [2] **and Abdullah Cahit Karaoglanli** [1]

1 Department of Metallurgical and Materials Engineering, Faculty of Engineering, Architecture and Design, 74110 Bartin, Turkey; karaoglanli@bartin.edu.tr
2 Department of Mechanical Engineering, Faculty of Engineering, Architecture and Design, 74110 Bartin, Turkey; dervisozkan@bartin.edu.tr (D.O.); msabrigok@bartin.edu.tr (M.S.G.)
* Correspondence: mehmetkilic307@hotmail.com; Tel.: +90-544-3296-347

**Abstract:** In this study, CoNiCrAlY metallic coatings were deposited on an Inconel 718 nickel-based superalloy substrate material using the detonation gun (D-gun) and supersonic plasma spraying (SSPS) techniques. The microstructural and mechanical properties in addition to their room and high-temperature wear behavior of the produced coatings were evaluated. The wear tests were performed at room temperature (rt), 250 and 500 °C using 2N and X-ray diffraction (XRD), scanning electron microscopy (SEM), and energy-dispersive spectroscopy (EDS) analyses of the worn coatings were performed to assess their wear performance. The coatings produced with D-gun process exhibited higher hardness and lower porosity ($550 \pm 50$ HV$_{0.25}$ hardness and $1.2 \pm 1.0\%$ porosity) than SSPS coatings (with $380 \pm 30$ HV$_{0.25}$ hardness and $1.5 \pm 1.0\%$ porosity) which resulted in better room- and high temperature wear performance for D-gun coatings. The worn surfaces of both coatings exhibited formation of tribological layers and superficial microstructural changes by varying temperature and load conditions. Increasing load and temperature resulted in increased wear loss whereas increasing temperature resulted in reduced COF values for both coatings.

**Keywords:** high temperature wear behavior; dry sliding wear; CoNiCrAlY; detonation gun (D-gun); supersonic plasma spraying (SSPS)

## 1. Introduction

Thermal spray coatings are widely used as overlay coatings to improve the wear, oxidation and corrosion resistance of engineering materials as a surface modification technique [1,2]. There are many areas of usage notably aviation and defense industry, automotive, mechanics, chemistry and many others to get wear, oxidation, thermal shock and corrosion resistance [3–7]. Literature showed that the generally used thermal spray techniques for depositing metallic bond coatings are atmospheric plasma spraying (APS), vacuum plasma spraying (VPS), high-velocity oxygen fuel (HVOF) and cold gas dynamic spraying (CGDS) [8–11].

MCrAlY metallic materials have been developed by considering high-temperature damage like oxidation and corrosion to protect the substrate material [12]. An MCrAlY coating system (M is Ni, Co or combination both elements) is widely used for turbine blades and vanes of gas turbine engines to provide surface protection [13,14]. MCrAlY-type coatings are widely applied in the form of NiCrAlY, NiCoCrAlY and CoNiCrAlY on nickel-based superalloy substrates. CoNiCrAlY alloys are commonly used as a metallic bond coating layer in thermal barrier coatings (TBCs) due to its

excellent high-temperature strength [15,16]. TBCs are protective coating systems on the purpose of thermal insulation in gas turbine engine blades and vanes in the aviation industry [17]. Generally, a TBC system consists of a metallic bond coat (MCrAlY; M = Co, Ni or Co/Ni), a ceramic top coat (YSZ, yttria-stabilized zirconia), and a thermally grown oxide layer (TGO). The ceramic top coat has an essentially low thermal conductivity and low thermal expansion and the metallic bond coat is deposited between the metallic substrate and ceramic top coat for developing the adherence of the ceramic top coating to the substrate alloy [9,16,18]. The basic purpose of CoNiCrAlY layer is the adhesion of ceramic top coating with base surface and these alloys help chemical stability which ceramic top coating cannot provide due to the porous surface [19].

Improving high-performance wear-resistant coating materials to protect the metallic substrate is an efficient approach to reducing wear. MCrAlY overlay coating is used as a protective coating against high-temperature oxidation and corrosion. Due to the sufficient amount of Al in the coatings during service life is the main degradation factor in conventional MCrAlY coatings. The performance of a metallic bond coat can be improved by applying Al and Cr gradient MCrAlY coatings [20]. Plasma spray, HVOF and CGDS coating techniques are used producing of CoNiCrAlY containing bond coating structure [21]. Materials processed by the detonation gun (D-gun) and supersonic plasma spraying (SSPS) techniques are used to display alternative characteristics/qualities for the production of coatings. The SSPS process enables the production of coatings cheaper, faster and easier. The D-gun technique also enables the production of fast and denser coating structure [22]. It's expected from CoNiCrAlY metallic bond coating structure to be dense, to have low porosity and oxide due to primary role on TGO forming and growing [23]. D-gun spraying is a thermal spray coating process that creates a coating surface with extremely hard, wear resistance, good adhesion strength, dense structure, low porosity and compressive residual stresses [24–26]. Additionally, due to D-gun spraying being an intermittent coating process, very few pores and blanks appear in the sprayed coatings [27]. It is preferred for protecting gas turbine engine parts and increasing material life against various types of wear at high temperatures. Unlike the D-gun technique, SSPS technique enables the production of coatings more efficiently, faster and more easily with a high deposition rate on different substrates [28]. SSPS technique enables the production of more dense and high bonding strength coatings due to the shorter exposure of atmospheric conditions and faster production compared to the conventional plasma spray coating process [29]. SSPS's density, porosity and adhesion strength of the coating is also higher than the coating prepared by APS [30,31]. Compared to the APS technique, it is faster and more efficient in completely melting SSPS powders [24].

Thermal spray coatings have been widely utilized in various industrial applications against surface damages such as wear corrosion and oxidation. Therefore, low porosity and good adhesion behavior are desired for the coating. Thermally sprayed MCrAlY bond coats, such as the one used in the present research, are primarily used for serving as an overlay metallic bond coat for thermal barrier coating systems against damages such as high-temperature oxidation and hot corrosion. However, few studies are encountered related to their wear performance, particularly those deposited with D-gun and SPSS techniques. This work therefore aims to investigate the microstructural characteristics and wear behaviors of CoNiCrAlY-based metallic bond coats onto nickel-based superalloy substrate Inconel 718 using the D-gun and SSPS coating techniques. To analyze the high-temperature wear behavior of the coatings, wear tests were applied at different temperatures and different loads. Before and after high-temperature wear tests microstructural characteristics and mechanical properties of CoNiCrAlY coatings were examined. SEM, EDS, hardness and 3D topography of the produced and worn samples were comparatively evaluated. As a result of the study, it has been understood that the high-temperature wear performances of the coatings vary depending on the technique used in the production of coatings and their microstructural characteristics.

## 2. Materials and Methods

Inconel 718 nickel-based superalloy disc samples having 25.4 mm diameter and 5 mm height were used as the substrate material. Before the coating process, surface cleaning and grit blasting of the substrate samples were fulfilled. Grit blasting was applied under a 75° angle and 2.5 bar pressure conditions when the shotgun and the surface of the distance of the substrate was approximately 10 cm. Different process parameters have been used in the production of CoNiCrAlY metallic bond coatings with SSPS and D-gun coating techniques. As for the metallic bond coats, the D-gun (Perun-S, Kiev, Ukraine) and the SSPS (Kiev-S plasma installation, Kiev, Ukraine) techniques were applied. The thickness of metallic coatings with MCrAlY content is approximately 100 μm. Coating parameters of D-gun and SSPS coating techniques used in CoNiCrAlY bond coating production are given in Tables 1 and 2.

**Table 1.** Deposition parameters of CoNiCrAlY sprayed using the D-gun technique.

| Combustion Gas | Air Flow Velocity | Number of Shots | Spray Time | Spray Distance |
|---|---|---|---|---|
| $C_3H_8$ (7.5 slpm) $O_2$ (25 slpm) Air (5 slpm) | 6.7–15 slpm | 100 | 14 s | 110–150 mm |

**Table 2.** Deposition parameters of CoNiCrAlY sprayed using the SSPS technique.

| Airflow Velocity | Current | Voltage | Spray Distance |
|---|---|---|---|
| 715 slpm | 270 A | 380 V | 200 mm |

CoNiCrAlY coated substrates after wear tests by using D-gun and SSPS techniques. In the study, wear tests were performed at different loads (2 N and 5 N) and different temperatures (room temperature (rt), 250 and 500 °C) for each sample. Using the contact stylus instrument Hommelwerke device (Hommelwerke GmbH, Villingen-Schwenningen, Germany), the surface roughness values were obtained by measuring from 5 to 10 different points on each sample according to the standard. Microhardness measurements of materials and coatings were made in Duramin brand test device using 250 g load ($HV_{0.25}$) and 15 s dwell time. Porosity measurements of the coatings were made by defining the matrix and porosity structures in microstructures in the Image J image analysis program (Lucia), by taking the average of the measurements on 5 SEM images at 2500× magnification.

Dry-sliding wear tests were performed using a Pin-on-disc wear test device (Turkyus Brand, İstanbul, Turkey) with tungsten carbide (WC) balls of 6 mm diameter (Redhill, Prague, Czech Republic) at three different temperatures: rt, 250 and 500 °C. The hardness value of the hard carbide balls used in the experiments is 19 GPa. The wear tests of the coated samples were performed at 0.18 m/s sliding speed by applying 2 and 5 N loads. After the dry sliding, wear tests were performed under different loads on Pin-on-disc wear device, volume losses of samples were determined with the help of a 3D-profilometer (Huvitz, Gunpo, Korea) by taking the average of 10 microstructural area measurements and multiplying this with the track diameter. Coefficient of friction values were evaluated by the frictional force data received from the load cell of the wear test rig.

## 3. Results and Discussion

Figure 1 shows the 2D, 3D and cross-sectional views of the wear tracks. As shown in the figure, the track widths are not uniform along the wear tracks which necessitated performing numerous cross-sectional area measurements.

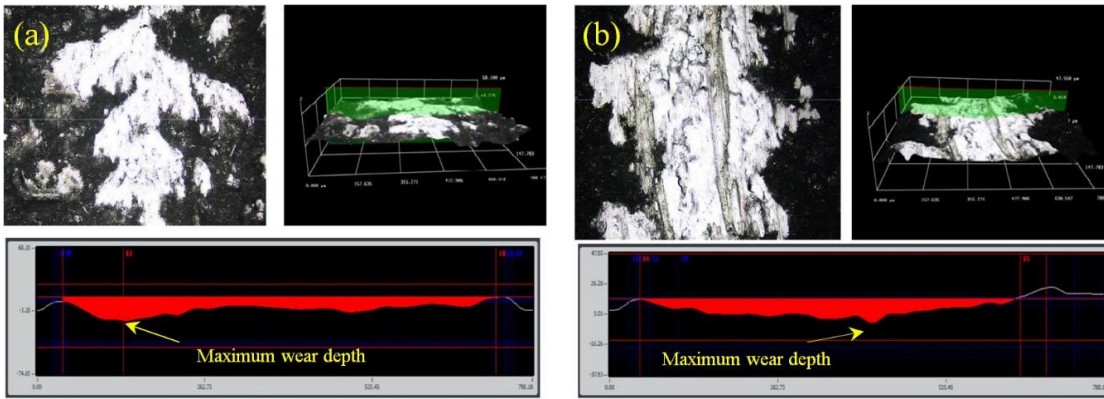

**Figure 1.** Scanning electron microscope (SEM) and 3D-profilometer image of the wear tracks: (**a**) produced by D-gun technique; (**b**) produced by the supersonic plasma spraying (SSPS) technique.

The hardness, surface roughness, porosity and oxide content parameters of CoNiCrAlY containing metallic bond coatings produced with D-gun and SSPS coating techniques are given in Table 3. It was determined that CoNiCrAlY metallic bond coatings produced by using the D-gun coating technique have higher hardness than the metallic bond coatings produced by SSPS coating technique. The reason for this is that the particles forming the coatings undergo plastic deformation which provides a dense coating structure [32].

**Table 3.** Hardness, surface roughness, porosity and oxide content of CoNiCrAlY coatings produced with the D-gun and SSPS techniques.

| Coating Method | Hardness ($HV_{0.25}$) | $R_a$ (μm) | Porosity (%) | Oxide (%) |
|---|---|---|---|---|
| D-gun | 550 ± 50 | 4.50 | 1.2 ± 1.0 | 29 ± 3.0 |
| SSPS | 380 ± 30 | 6.90 | 1.5 ± 1.0 | 9 ± 2.0 |

The cross-sectional SEM micrographs of the as-sprayed CoNiCrAlY coatings are shown in Figure 2. These coating microstructures have high porosity content in addition to oxide formations. In the figure, regions with porosity are observed in black color, and those with oxide content are observed in dark and light grey.

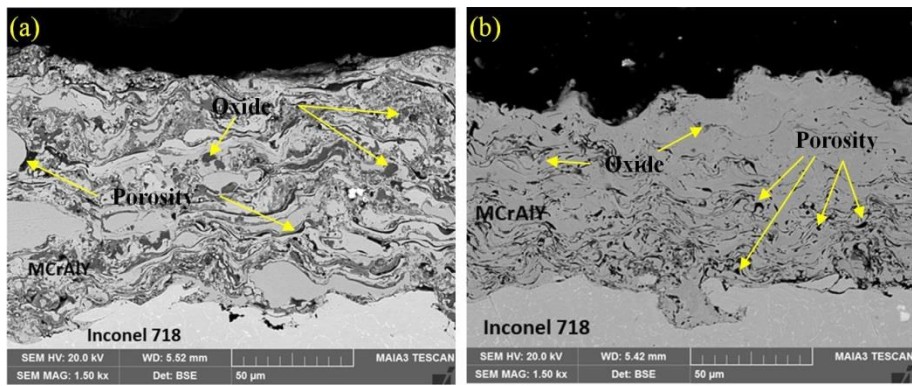

**Figure 2.** Cross-sectional SEM image of as-sprayed CoNiCrAlY coatings: (**a**) D-gun technique; (**b**) SSPS technique.

Figure 2a shows the microstructure of the coating deposited by the D-gun technique. The differences in the porosity and oxide content of the coatings produced by SSPS and D-gun techniques can be seen in SEM microstructures. The as-sprayed SSPS-CoNiCrAlY coatings exhibited a porous microstructure having micro-cracks as expected from a typical plasma spraying microstructure.

It is seen from the SEM images that the surface roughness of the coating produced by the SSPS technique is lower compared to those produced by the D-gun technique. This situation occurs depending on the production characteristics of the coatings and the production conditions [3,26,27]. It causes damage formation that result in a lower level of oxidation resistance and detaching of the coating from the surface [33].

After the dry sliding wear tests performed at different temperatures and different loads, the volume loss tables were obtained with the 3D-profilometer, as shown in Table 4. Generally, the samples produced by the D-gun technique showed better wear resistance both at room temperature and high temperatures (250 and 500 °C) [7]. It is possible to associate the first reason for the hardness. The other effective parameter is microstructural features that come from the production process and conditions. Hardness values of coatings produced by using the D-gun technique are comparatively higher than the other thermal spray coating techniques [34]. When the coatings produced by D-gun and SSPS techniques were evaluated microstructurally, it is seen from the microstructures that the coatings produced by the SSPS technique have higher porosity content. In the D-gun coating technique, denser coating structure is obtained because of the high speed used in the production of coatings [32]. The microstructural defects, high porosity structure of SSPS coatings have had a negative effect on the wear, strength and toughness properties of the coatings. The techniques used in the production of coatings are mainly effective in their performance during usage conditions [35].

**Table 4.** Volume losses of the samples at varying load and temperature conditions of D-gun and SSPS coating technique. Room temperature—rt.

| Temperature (°C) | D-gun | | SSPS | |
|---|---|---|---|---|
| | Load | | | |
| | 5 N | 2 N | 5 N | 2 N |
| rt | 0.47 | 0.45 | 0.44 | 0.77 |
| 250 °C | 0.39 | 0.56 | 0.38 | 0.57 |
| 500 °C | 0.25 | 0.52 | 0.29 | 0.48 |

Table 4 shows that a decrease in volume loss occurred in almost all samples with increasing load. The reason for this may be related to the fact that the oxide layer formed on the surface; this formed oxide layer becomes more composite under heavy load and increases the wear resistance [36]. With the start of the wear process that is, with the abrasive ball and the relative movement of the opposite surface the surface roughness will break first. This roughness that is broken at the micron level, some of which remain on the surface of the material as wear debris. With the movement and increased pressure of the abrasive ball on them, they will both be oxidized and buried in the lower surface. Thus, the oxide layer formed on the surface will play an effective role in increasing wear resistance. As mentioned earlier, when the wear graph of the samples produced with the SSPS technique was examined under the same conditions, compared to the D-gun technique, higher volume loss was observed. However, with the increase of the load here, the volume loss values in the samples also increased. While the increase in the previous sample causes the volume losses to decrease, the opposite situation will be tried to relate to the microstructure and mechanical properties of the produced sample. It is briefly explained above how to decrease volume loss with increasing load for D-gun coating samples. A similar mechanism (breaking the roughness on the surface, composing, burying on the surface) is also valid here. When the hardness values of the samples are examined, it is seen that the hardness value of the samples produced by the SSPS technique is lower. Accordingly, the abrasive ball will have a deeper depth of penetration in the opposite surface under the same load. Therefore, the volume loss of the opposite surface material gathered in front of the cutter will increase, so the volume loss will increase. Furthermore, an increase of the failure with the increase of the load could be related to weakness of the bond strength between the splat of the samples produced by the SSPS coating technique than those produced with the D-gun technique [37].

At the end of the dry sliding wear tests, when the SEM wear track photos taken from the sample surfaces after being subjected to etching at rt under 2 N load are examined, it was found that the samples produced by both techniques are first exposed to oxidation wear. However, in the samples produced with the D-gun technique, it is observed that this occurs more severely in the SSPS technique, where the fatigue occurring under repeated loads is less effective. In the SSPS technique, micro-cracks that occur due to fatigue perpendicular to the direction of etching caused surface spalling in the future. In experiments carried out at rt under 5 N load, delamination type wear mechanism is seen in addition to previous wear mechanisms in the D-gun technique. In the SSPS technique, wear due to fatigue and spalling was detected.

At the SEM wear track photos obtained as a result of the experiments carried out at 250 °C at 2 N load (Figure 3), a more homogeneous oxide layer is observed on the surface of the samples treated with D-gun. It is also understood that this oxide layer is damaged by fatigue micro cracks. In the SSPS sample, it is seen that the oxide formation in the wear track region is not homogeneous. The reason for this can be explained by the fact that the sample causes more roughness, as the abrasive ball is starting to break the high roughness first. While some of these broken roughness's are removed from the system, some of them will remain between the abrasive and the substrate and will be crushed and become more compact. This is understood from the SEM photograph (Figure 3b). When the SEM microstructures images are examined (Figures 4 and 5), it is seen that there are spallings and breakings in parallel with the track of wear. This explains that the sample is not damaged by fatigue; it is dependent on the gaps underneath.

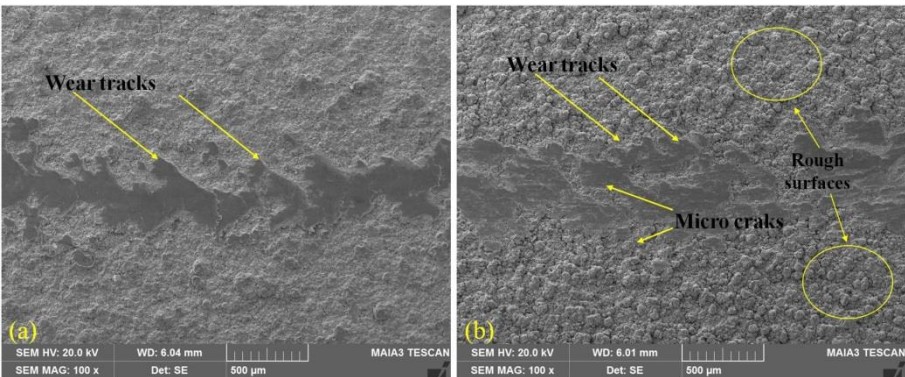

**Figure 3.** SEM micrographs after wear test at 250 °C under 2 N load: (**a**) D-gun coating technique; (**b**) SSPS coating technique.

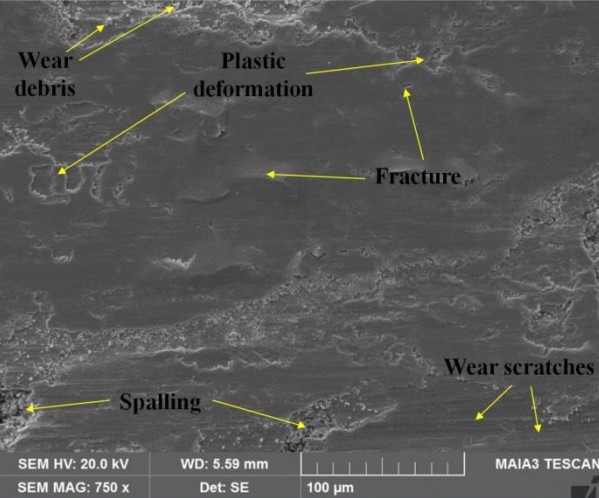

**Figure 4.** SEM micrograph of the worn surface of SSPS coating technique at 250 °C under 2 N load.

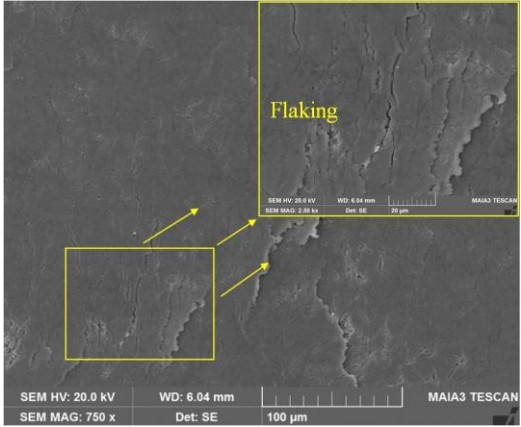

**Figure 5.** SEM micrograph of the worn surface of the D-gun coating technique at 250 °C under 2 N load.

When the SEM photos of the samples at 500 °C and 2 N load are examined, especially the technique produced by D-gun has much less volume loss. That is to say, when Figure 6a is examined, it is understood that the abrasive ball is in contact with a small portion of the surface. We can explain this situation as follows; the increased temperature will cause some elements in the material to oxidize much faster [3,9]. Depending on the nature of this oxidation, the oxide layer will either spalling quickly or steadily stick to the surface [30,34].

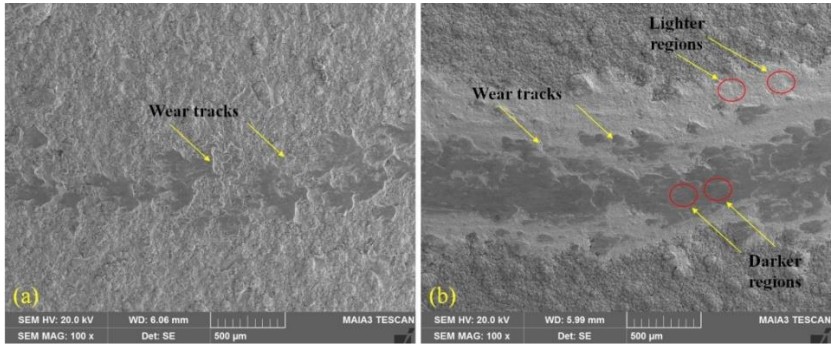

**Figure 6.** SEM micrographs after wear test of samples at 500 °C under 2 N load: (**a**) D-gun coating technique; (**b**) SSPS coating technique.

In this micrograph, it is understood that the oxide layer becomes compact under load and does not spall. As a result, the wear resistance of the material increased. However, the sample produced by the SSPS technique under 2 N loads conditions did not show a similar situation and it can be shown in Figure 6b. When the SEM photo is examined, two different wear areas attract attention. The region with a darker tone close to the center of the wear track and two adjacent regions with a lighter tone constitute the worn areas.

In Figures 7 and 8, when the linear and point EDS analysis from different regions of the wear surface are examined after the wear test under 500 °C and 2 N load is examined, it is understood that this ratio is low in the other region where the dark region is rich in Ni, Co and Al. The peaks here are thought to increase due to the compounds formed by the combination of elements and oxygen in the coating powder. This is because, when the point analysis given in the other tab of the same figure is analyzed, it can be seen that this decreases in the white region of 119 where spectrum 117 and 118 have more oxygen. The presence of Wolfram element, which is not present in the substrate, is remarkable in point analysis. With increasing temperature, it is understood that both the formation of darker oxide layers on the abraded surface and material transfer from the abrasive ball to the opposite surface due to the change in the plastic deformability of the abrasive wolfram ball, depending on the temperature. However, after the experiments carried out at rt and 250 °C, no tracks of tungsten were found on the

surface of the material in the EDS analysis results taken in the wear track region. The friction coefficient and microstructural changes occurring on the wearing surfaces at temperature and pressure changes and varying according to time are examined graphically. While there was a continuous increase in rt under 2 N load depending on time, it was not observed that there was a multidirectional increase in temperatures of 250 and 500 °C. It was observed that the highest coefficient of friction was at rt and depending on the increase in temperature, the microstructures of the samples are deteriorated.

The friction coefficient values of the samples coated with the D-gun technique are given in Figure 9a. While the lowest friction coefficient data were initially obtained at under rt and 2 N load, this value reached the friction coefficient values at 250 and 500 °C after 50 s. The friction coefficient value obtained in the wear process performed at 500 °C was lower than that at 250 °C. The reason for this may be associated with the increase in the thickness and speed of formation of the oxide layer formed on the surface with the increase in temperature [3,9].

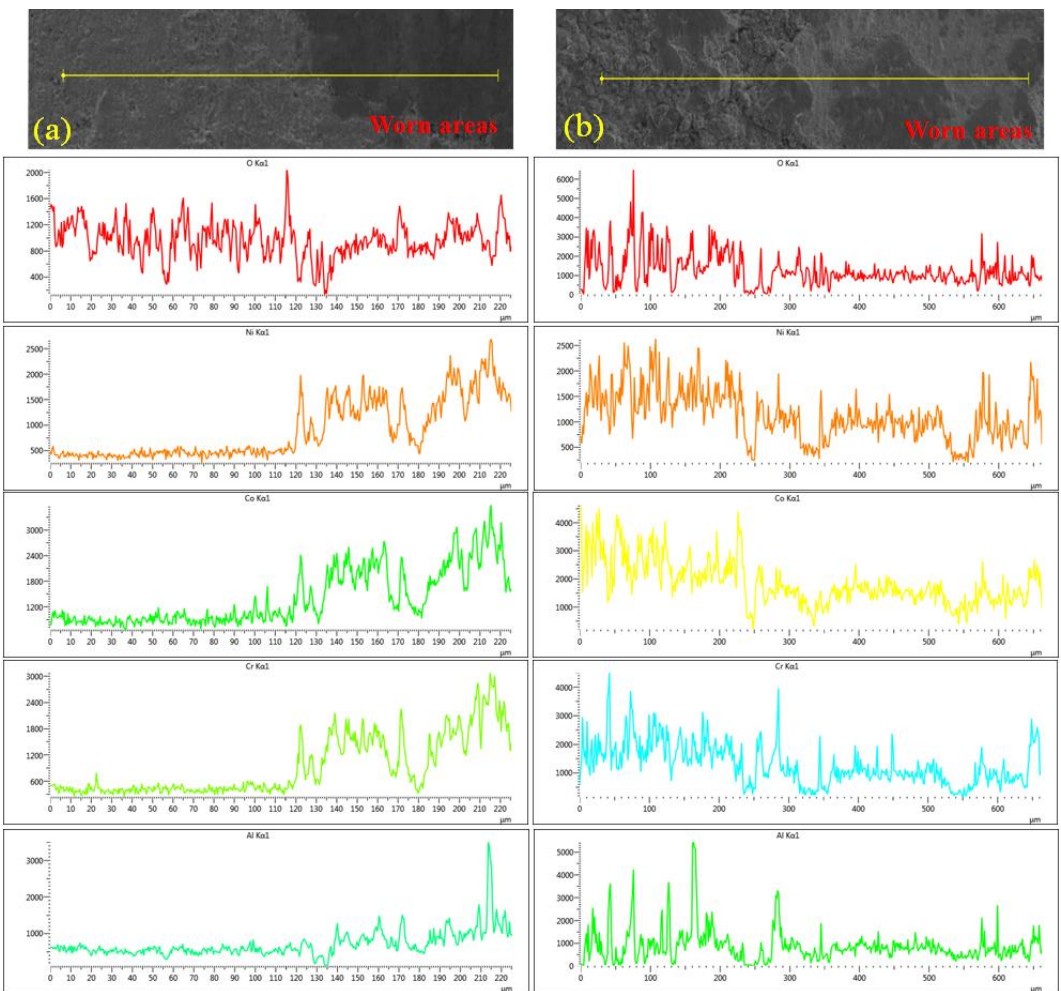

**Figure 7.** EDS linear analysis of wear tracks at 500 °C under 2 N load: (**a**) D-gun coating technique; (**b**) SSPS coating technique.

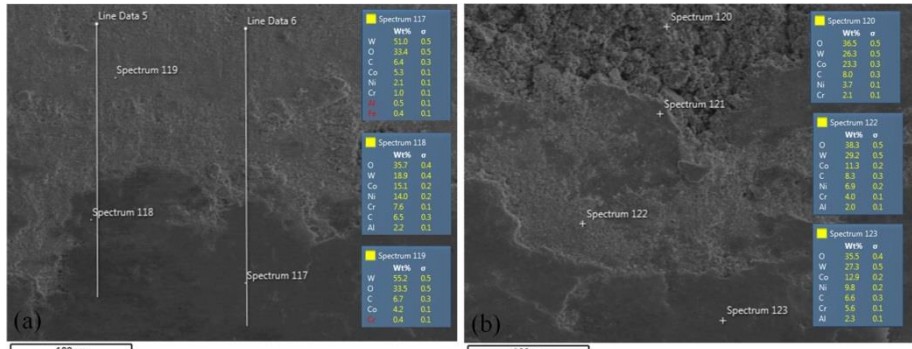

**Figure 8.** Surface micrographs with EDS point analysis of worn surfaces at 500 °C under 2 N load: (**a**) D-gun coating technique; (**b**) SSPS coating technique.

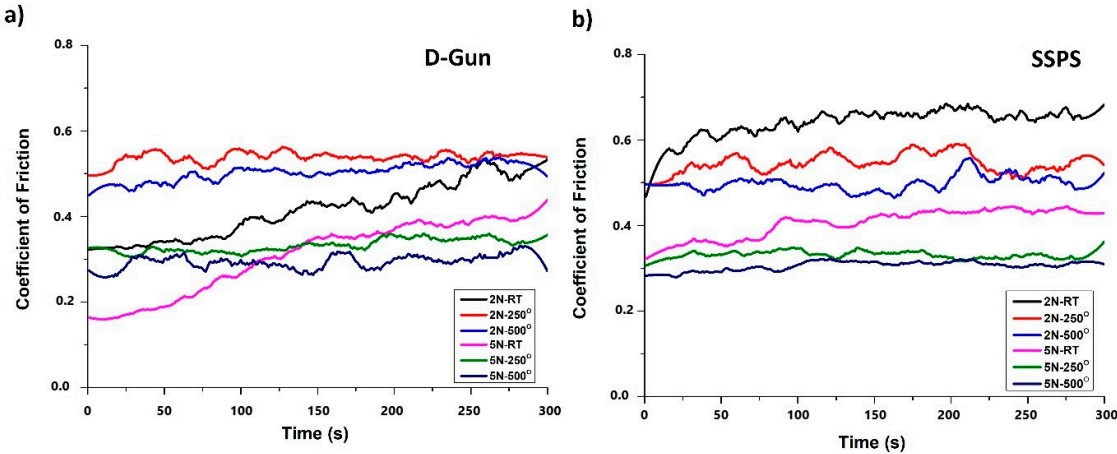

**Figure 9.** Coefficient of friction curves of the coatings at different temperatures under 2 and 5 N loads: (**a**) D-gun coating technique; (**b**) SSPS coating technique.

A similar situation was observed after the experiments carried out under 5 N load in the friction graphs. However, the friction coefficient values of these samples were at lower levels. The friction coefficient values of the samples coated with the SSPS technique are given in Figure 9b. The lowest friction coefficients were determined at 5 N load and 500 °C. This was followed by 250 °C and rt temperatures. When the friction coefficients were examined under 2 N load, the lowest value was observed in the sample at 500 °C. This was followed by 250 °C and rt values. However, when the samples processed under 2 and 5 N loads are examined (Figure 9b), it is noteworthy that the fluctuations in the friction coefficient graphs in 2 N are higher and the graphs are more stable in 5 N loads. It can be attributed that these fluctuations under 2 N load occur as a result of spalling the oxide layer formed faster than the surface.

## 4. Conclusions

In this study, metallic bond coatings with CoNiCrAlY content were produced on Inconel 718 superalloy substrate material by using D-gun and SSPS coating techniques. The coatings produced were subjected to wear tests at three different temperatures: rt, 250 and 500 °C and different loads 2 and 5 N to understand the wear mechanisms. The salient conclusions arising from this work are as follows:

1. CoNiCrAlY metallic powders were successfully deposited on the Inconel 718 superalloy substrates using D-gun and SSPS coating techniques.
2. The high-temperature wear behavior of the coatings has changed depending on the processes used in the coating production and the microstructural properties of the coatings after production.



3.  Depending on the increasing loading rates and temperature, wear losses were likewise increased. However, this increase was not linear.

4.  For D-gun and SSPS coatings, increasing load resulted in lower coefficient of friction values. Increasing temperature resulted in lower COF values for SSPS coatings; however, those produced with D-gun did not follow the same trend.

5.  It has been understood that at 250 °C and rt surface fatigue wear by using D-gun technique is comparatively more severe than SSPS technique.

6.  It has been observed that tribological layers and superficial changes occur in the microstructures of the coatings due to temperature and time by both thermal spray coating techniques.

7.  When high-temperature wear behaviors of CoNiCrAlY coatings are examined, it is seen that D-gun coatings show superior properties compared to SSPS coatings.

8.  Due to the increased surface hardness and microstructural dense structure with high-temperature effect, the wear resistance of the coatings increases.

**Author Contributions:** A.C.K. and D.O. designed the experiments; A.C.K. and D.O. performed the experiments; A.C.K., D.O., and M.S.G. analyzed the data; A.C.K., D.O., M.S.G., and M.K. wrote, reviewed and edited the paper. All authors have read and agreed to the published version of the manuscript.

**Funding:** This research received no external funding.

**Acknowledgments:** The authors would like to acknowledge Vitova Ltd. for the production of detonation gun (D-gun) and supersonic plasma spraying (SSPS) CoNiCrAlY samples.

**Conflicts of Interest:** The authors declare no conflict of interest.

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
