# Peer review of "Room- and high temperature Wear Resistance of MCrAlY Coatings Deposited by Detonation Gun (D-gun) and Supersonic Plasma Spraying (SSPS) Techniques"

_coatings, doi:10.3390/coatings10111107_

Round 1

Reviewer 1 Report

The manuscript compared the anti-abrasive performances of two technicals used for MCrAlY coatings at elevated temperature environment.

The manuscript could not be accepted for publication in its present form.

1) The abstract should be re-written. More detals on the perfromance, quantified data, mechanism should be illustrated.

2) The introduction part is in a mess and in logical disorder. The purpose of the research work also should be explained with more soundness reason. The reviewer is confused: Is MCrAlY coatings used alone? Is the anti-abrasion performance important?  In the opinion of the reviewer, the abrasion is occur on the surface of the YSZ, instead of on the surface of the MCrAlY.

3) Line 117 -121 is a result of the experiments, should not be in the experimental part. The "hardness" should be "harder"?

4) In Figure 1, only (a) and (b) is labelled and explained for the sub-title.

5) In Figure 3, it is hard to compare the volume among samples. could the author use another figure formation?

6) There are many grammar erro in the text labelled with green color.

Author Response

Response to Reviewer 1 Comments

Dear Reviewer ,

The authors thank to Reviewer 1 to his/her valuable contributions. We are also grateful for the positive evaluations of Reviewer 1.

Point 1: The abstract should be re-written. More detals on the perfromance, quantified data, mechanism should be illustrated.

Response 1: Abstract section is rewritten in accordance with the reviewer comments as follows:

In this study, CoNiCrAlY metallic coatings were deposited on an Inconel 718 Nickel-based superalloy substrate material using the detonation gun (D-gun) and supersonic plasma spraying (SSPS) techniques. The microstructural and mechanical properties in addition to their room and high temperature wear behavior of the produced coatings were evaluated. The wear tests were performed at room temperature (rt), 250°C and 500°C using 2N and X-ray diffraction (XRD), scanning electron microscopy (SEM), and energy-dispersive spectroscopy (EDS) analyses of the worn coatings were performed to assess their wear performance. The coatings produced with D-gun process exhibited higher hardness and lower porosity (550 ± 50 HV0.25 hardness and 1.2±1.0 % porosity) than SSPS coatings (with 380 ± 30 HV0.25 hardness and 1.5±1.0 % porosity) which resulted in better room and high temperature wear performance for D-Gun coatings. The worn surfaces of both coatings exhibited formation of tribological layers and superficial microstructural changes by varying temperature and load conditions. Increasing load and temperature resulted in increased wear loss whereas increasing temperature resulted in reduced COF values for both coatings. 

Point 2: The introduction part is in a mess and in logical disorder. The purpose of the research work also should be explained with more soundness reason. The reviewer is confused: Is MCrAlY coatings used alone? Is the anti-abrasion performance important?  In the opinion of the reviewer, the abrasion is occur on the surface of the YSZ, instead of on the surface of the MCrAlY.

Response 2: The authors are grateful for the positive contribution of the reviewer. The introduction part is now modified, information related to abrasion is removed from the study, as only dry sliding room and high temperature wear behavior of the coatings is investigated. The title of the research is also modified accordingly. MCrAlY coating was applied as a single coating layer as opposed to TBC systems. The following statement is included in the last paragraph of the introduction section to avoid confusion for readers:

Thermally sprayed MCrAlY bond coats, such as the one used in the present research, are primarily used for serving as an overlay metallic bond coat for thermal barrier coating systems against damages such as high temperature oxidation and hot corrosion. However, few studies are encountered related to their wear performance, particularly those deposited with D-gun and SPSS techniques. 

Point 3: Line 117-121 is a result of the experiments, should not be in the experimental part. The "hardness" should be "harder"?

Response 3: The required correction is made. The sentence is corrected and moved to the results and discussion section.

Point 4: In Figure 1, only (a) and (b) is labelled and explained for the sub-title.

Response 4: Related discussion is included in the manuscript for Figure 1.

Point 5: In Figure 3, it is hard to compare the volume among samples. could the author use another figure formation?

Response 5: The figure is replaced with a Table including all the worn volume results.

Point 6: There are many grammar erro in the text labelled with green color.

Response 6: Thanks to the efforts and attention of the reviewer, the highlighted grammar errors are now corrected and highlighted with blue font.

Reviewer 2 Report

The article investigates the wear behaviors of CoCrAlY coatings prepared by D-gun and SSPS. he article needs to be improved before publication. 

  • According to the introduction, SSPS has an advantage in making dense layer with higher production, however, authors compare wear result of coating materials by D-gun and SSPS only. Why did you compare the wear behavior of D-gun and SSPS? Isn't there any data of APS, VPS or HVOF? I suggest they should be compare with any commercial coating method (If they don't, it needs to be discussed basing on the reference, at least). 
  • In this work, TBC was never used, then, why the TBC is mentioned in the introduction?
  • Move Table 3 and Fig. 1 to the result section.
  • Illustrate how you analyse hardness, roughness, and oxide content in Table 3 and present in the section 2.
  • Present the material of ball indenter exactly (maker? tungsten carbide?)
  • How did you evaluate the volume loss in the wear test?
  • I cannot tell the color difference in Fig. 10. 
  • What's the clear conclusion concerning about the effect of load or temperature? 

Author Response

Response to Reviewer 2 Comments

Dear Reviewer ,

The authors thank to Reviewer 2 to his/her valuable contributions. We are also grateful for the positive evaluations of Reviewer 2.

Point 1: According to the introduction, SSPS has an advantage in making dense layer with higher production, however, authors compare wear result of coating materials by D-gun and SSPS only. Why did you compare the wear behavior of D-gun and SSPS? Isn't there any data of APS, VPS or HVOF? I suggest they should be compare with any commercial coating method (If they don't, it needs to be discussed basing on the reference, at least). 

Response 1: Numerous studies on HVOF, VPS and APS deposited MCrAlY coatings are available in the literature, therefore we decided to determine the wear behavior of MCrAlY coatings deposited with D-gun and SPSS and this is now mentioned in the last paragraph of the introduction section.  

Point 2: In this work, TBC was never used, then, why the TBC is mentioned in the introduction?

Response 2: As you pointed out, no TBC is used in the performed study, however the primary function of MCrAlY coatings is to serve as a bonding layer between the substrate and top coating layer of TBC systems, therefore we needed to mention its role in thermal barrier coating systems. We here investigate an alternative usage for MCrAlY coatings (protection against wear), and this is now highlighted in the manuscript with the following statement in the last paragraph of the introduction section:

 “Thermally sprayed MCrAlY bond coats, such as the one used in the present research, are primarily used for serving as an overlay metallic bond coat for thermal barrier coating systems against damages such as high temperature oxidation and hot corrosion. However, few studies are encountered related to their wear performance, particularly those deposited with D-gun and SPSS techniques.”

Point 3: Move Table 3 and Fig. 1 to the result section.

Response 3: They are now moved to the Results and Discussion section along with relevant discussion in the manuscript.

Point 4: Illustrate how you analyse hardness, roughness, and oxide content in Table 3 and present in the section 2.

Response 4: Further details related to the microhardness, roughness and oxide content measurements are now included in Section 2 as follows:

Using the contact stylus instrument Hommelwerke device, the surface roughness values were obtained by measuring from 5 to 10 different points on each sample according to the standard. Microhardness measurements of materials and coatings were made in Duramin brand test device using 250 g load (HV0.25) and 15 seconds dwell time. Porosity measurements of the coatings were made by defining the matrix and porosity structures in microstructures in the Image J image analysis program, by taking the average of the measurements on 5 SEM images at 2500x magnification.

Point 5: Present the material of ball indenter exactly (maker? tungsten carbide?)

Response 5: Information related to the producer and material of the ball indenter are now included in the manuscript with the following statement:

Dry-sliding wear tests were performed using a Pin-on-disc wear test device (Turkyus Brand-Turkey) with tungsten carbide (WC) balls of 6 mm diameter (Redhill-Czech Republic) at three different temperatures: rt, 250°C and 500°C.

Point 6: How did you evaluate the volume loss in the wear test?

Response 6: The relevant information is now included in the manuscript as follows:

After the dry sliding wear tests were performed under different loads on Pin-on-disc wear device, volume losses of samples were determined with the help of a 3D-profilometer (Huvitz-Republic of Korea) by taking the average of 10 microstructural area measurements and multiplying this with the track diameter. Coefficient of friction values were evaluated by the frictional force data received from the load cell of the wear test rig.

Point 7: I cannot tell the color difference in Fig. 10. 

Response 7: Figure 10 is now replaced with a modified version of the COF graphs. Bold font was used for all text  and all lines were thickened.

Point 8: What's the clear conclusion concerning about the effect of load or temperature? 

Response 8: The following statement is included in the Conclusions section:

For D-gun and SSPS coatings, increasing load resulted in lower coefficient of friction values. Increasing temperature resulted in lower COF values for SSPS coatings, however those produced with D-Gun did not follow the same trend.

Round 2

Reviewer 1 Report

The manuscript was revised extensively and can be accepted for publication.

Reviewer 2 Report

The paper is now acceptable as they revised according to reviewer's comments. 

This manuscript is a resubmission of an earlier submission. The following is a list of the peer review reports and author responses from that submission.

Round 1

Reviewer 1 Report

This paper discussed the investigations of microabrasion, dry sliding and wear resistance of MCrAlY coatings deposited by the detonation gun and the supersonic plasma spraying. This paper has highly scientific values with comparing between the detonation gun and the supersonic plasma spraying.

(1) Please show the company name and model number of the used spray guns.

(2) In Table 3, “Roughness” would be modified to “hardness”.

(3) In conclusion “7. Depending on the increase of surface hardness in coating structures, wear resistance has also increased in coating structures produced using both techniques.”

 This sentence is not understandable. Is the increase of surface hardness meaning to the oxide layers were produced under the sliding tests, and these oxide layers were thickening by the increased temperature of the sliding test?

Author Response

Response to Reviewer 1 Comments

Dear Reviewer,

The authors thank Reviewer 1 to his/her valuable contributions. We are also grateful for the positive evaluations of Reviewer 1.

This paper discussed the investigations of microabrasion, dry sliding and wear resistance of MCrAlY coatings deposited by the detonation gun and the supersonic plasma spraying. This paper has highly scientific values with comparing between the detonation gun and the supersonic plasma spraying.

Point 1: Please show the company name and model number of the used spray guns.

Response 1: According to your comment, related information were added in the manuscript.

As for the metallic bond coats, the D-gun (Perun-S, Ukraine) and the SSPS (Kiev-S plasma installation, Ukraine) techniques were applied.

Point 2: In Table 3, “Roughness” would be modified to “hardness”.

Response 2: Thank you very much for your valuable attention. According to your comment, we have changed the wrong term in the manuscript.

Table 3. Hardness, surface roughness, porosity and oxide content of CoNiCrAlY coatings produced with D-gun and SSPS techniques.

Coating method

Hardness (Hv)

Ra (µm)

Porosity (%)

Oxide (%)

D-Gun

550 ± 50

4.50

1.2±1.0

29±3.0

SSPS

380 ± 30

6.90

1.5±1.0

9±2.0

Point 3: In conclusion “7. Depending on the increase of surface hardness in coating structures, wear resistance has also increased in coating structures produced using both techniques.”

This sentence is not understandable. Is the increase of surface hardness meaning to the oxide layers were produced under the sliding tests, and these oxide layers were thickening by the increased temperature of the sliding test?

Response 3: According to your comment, the sentence was changed and rewrite this sentence in the manuscript.

Due to the increased surface hardness and microstructural dense structure with high-temperature effect, the wear resistance of the coatings increases.

Reviewer 2 Report

1、In the part of INTRODUCTION, without any information about the friction and wear properties of CoNiCrAlY or the purpose of this work.

2、Table 3 should be corrected.

3、Conclusions should be re-written. 

Author Response

Response to Reviewer 2 Comments

Dear Reviewer,

The authors thank Reviewer 2 to his/her valuable contributions. We are also grateful for the positive evaluations of Reviewer 2.

Point 1: In the part of INTRODUCTION, without any information about the friction and wear properties of CoNiCrAlY or the purpose of this work.

Response 1: According to your comments, we have added the information in the manuscript.

Improving the high-performance wear-resistant coating materials to protect the metallic substrate is an efficient approach to reducing wear. For this purpose, the wear rate and friction coefficient are significant parameters in the performance of the coated components subjected to contact with others at high temperature. Besides, wear performances are generally related to the surface hardness of the material. For instance, wear resistance is especially important at blade tips, which run into a worn layer on the inner turbine shroud for perfect sealing. For that reason, the MCrAlY coatings have been used as the protective APS coatings against frictional wear at higher temperatures [20-22].

Point 2: Table 3 should be corrected.

Response 2: According to your comment, Table 3 was corrected and added in the manuscript.

Table 3. Hardness, surface roughness, porosity and oxide content of CoNiCrAlY coatings produced with D-gun and SSPS techniques.

Coating method

Hardness (Hv)

Ra (µm)

Porosity (%)

Oxide (%)

D-gun

550 ± 50

4.50

1.2±1.0

29±3.0

SSPS

380 ± 30

6.90

1.5±1.0

9±2.0

Point 3: Conclusions should be re-written.

Response 3: According to your comment, conclusion part re-written and added in the manuscript.

In this study, metallic bond coatings with CoNiCrAlY content were produced on Inconel 718 superalloy substrate material by using D-gun and SSPS coating techniques. The coatings produced were subjected to wear tests at three different temperatures: rt, 250°C and 500°C and different loads 2N and 5N in order to understand the wear mechanisms. The salient conclusions arising from this work are as follows:

  1. CoNiCrAlY metallic coating powders were successfully deposited on the nickel-based Inconel 718 superalloy substrate using D-Gun and SSPS coating techniques.
  2. The high-temperature wear behavior of the coatings has changed depending on the processes used in the coating production and the microstructural properties of the coatings after production.
  3. Depending on the increasing loading rates and temperature, wear losses were likewise increased. However, this increase was not linear.
  4. It has been understood that at 250 °C and rt surface fatigue-wear by using D-gun technique are comparatively more severe than SSPS technique.
  5. It has been observed that tribological layers and superficial changes occur in the microstructures of the coatings due to temperature and time by both thermal spray coating techniques.
  6. When high-temperature wear behaviors of CoNiCrAlY coatings are examined, it is seen that D-gun coatings show superior properties compared to SSPS coatings.
  7. Due to the increased surface hardness and microstructural dense structure with high-temperature effect, the wear resistance of the coatings increases.

Reviewer 3 Report

Review of Manuscript

Microabrasion, dry sliding and high temperature  wear resistance of MCrAlY coatings deposited by detonation gun (D-gun) and supersonic plasma  spraying (SSPS) techniques

The topic is important and being discussed in literature widely. However, the task statement is unclear. It is difficult to understand the results because of poor English. The terminology being used is not correct in many cases. So, considerable rewriting and deeper analysis of experimental data is needed to eliminate mistakes and to be published

Some comments are below. It was impossible to deeply analyse all results and discussion because of difficulties of understanding.

  1. Line 24-25: The sentence “…The results show that the high  temperature wear resistance of the coatings has changed with the microstructural and surface  features of the samples depending on the coating production process and wear mechanisms” is not completely clear. About what surface features are you talking?  Please, specify  the new specific features of studied wear mechanisms in Abstract. English is not proper.
  2. Line 45-47: The sentence “…TBC systems are generally consisted from chemically stabile  superalloy base material, CoNiCrAIY content owner high oxidation and corrosion resistance  metallic bonding coating, porosity content owner heat insulating ceramic top coating and thermally  growing oxide layer (TGO) consisted in bond and top coating interface during operating conditions” is not correct and clear. What does it mean “…CoNiCrAIY content owner, …porosity content owner ”? “TBC  systems” do not consist of superalloy base material. It consists of Ceramics (ZrO2) and CoNiCrAIY bond layers.
  3. Line 49: Citing [17] is not relevant here. This paper is only  about shot peening.
  4. Line 55-56: What does it mean: “…Used for faster, easier, denser and more economical coating structure producing techniques are separated each other’s microstructural and  mechanical way [20]”?  English is not proper.
  5. Line 65-66: What does it mean “thinner microstructure“ in the sentence “…When compared to traditional plasma spray yttria stabilized zirconia (YSZ) coatings, SSPS coating has a thinner microstructure and higher  adhesion strength”? The sentence is not completed. Looks like you need to add: “…. as compared to that of….”
  6. Line 75-77: Unfortunately authors do not clearly set up the paper tasks. “High temperature wear resistance” is defined as a function of “coating production technique”, “microstructural properties” and “wear  mechanisms due to temperature and load”. Please, take a look at the sentence: “It has  been observed that the high temperature wear resistance of the coatings varies according to the  coating production technique, the microstructural properties of the coating, as well as the wear  mechanisms due to temperature and load”. It is completely not correct. What does it mean: “microstructural properties” ?
  7. Line 92: Please, put the scale on the images. Figure 1. Macro image of CoNiCrAlY coated substrates after wear tests: (a) produced by SSPS  technique; (b) produced by D-gun technique.  
  8. Lines1132-133: Unfortunately it is very difficult to understand: “…It can be seen from the microstructures that the CoNiCrAlY coating structure produced by the D-gun technique has a much more oxidized  structure and the internal porosity content is lower than the CoNiCrAlY coating structure produced  by the SSPS coating technique”. It means that “…internal porosity content is lower than the CoNiCrAlY coating structure”. How to compare porosity and structure????
  9. Lines 134-138: The similar sentence: “It is seen that the surface roughness of the coating structure produced by the SSPS coating technique has a lower roughness value compared to the process produced by  the D-gun coating technique. A higher level of oxide formation is the primary factor affecting the  oxide formation on interface depending on coating production technique, temperature and time of  experimental procedures [3,24,25]. It causes damage formation that result in lower level of oxidation  resistance and detaching of the coating from the surface [31]”. The unclear terms are followings: “surface roughness of the coating structure”, “…It is seen that the surface roughness…. has a lower roughness”, “lower level of oxidation  resistance”, “level… of detaching of the coating from the surface”.  
  10. Line 296-297 How to understand the sentence in conclusion “…As a result of the analysis, it was observed that when the coatings produced by SSPS technique were tested at rt temperature was separated from the surface owing to regional breaks and spalling, this structure formed after cleaning the particles on the surface and forming a compact structure”?
  11. Lines 139- 290. Unfortunately, the similar mistakes, unclear statements and terminology mistakes are in all manuscript chapters.

Author Response

Response to Reviewer 3 Comments

Dear Reviewer,

The authors thank Reviewer 3 to his/her valuable contributions. We are also grateful for the positive evaluations of Reviewer 3.

The topic is important and being discussed in literature widely. However, the task statement is unclear. It is difficult to understand the results because of poor English. The terminology being used is not correct in many cases. So, considerable rewriting and deeper analysis of experimental data is needed to eliminate mistakes and to be published.

Some comments are below. It was impossible to deeply analyse all results and discussion because of difficulties of understanding.

Point 1: Line 24-25: The sentence “…The results show that the high temperature wear resistance of the coatings has changed with the microstructural and surface features of the samples depending on the coating production process and wear mechanisms” is not completely clear. About what surface features are you talking? Please, specify the new specific features of studied wear mechanisms in Abstract. English is not proper.

Response 1: According to your comment, related abstract part re-written and added in the manuscript.

The high-temperature wear behavior of the coatings has changed depending on the processes used in the coating production and the microstructural properties of the coatings after production.

Point 2: Line 45-47: The sentence “…TBC systems are generally consisted from chemically stabile superalloy base material, CoNiCrAIY content owner high oxidation and corrosion resistance metallic bonding coating, porosity content owner heat insulating ceramic top coating and thermally growing oxide layer (TGO) consisted in bond and top coating interface during operating conditions” is not correct and clear. What does it mean “…CoNiCrAIY content owner, …porosity content owner ”? “TBC systems” do not consist of superalloy base material. It consists of Ceramics (ZrO2) and CoNiCrAIY bond layers.

Response 2: According to your comment, related introduction part re-written and added in the manuscript.

Generally, a TBC system consist of a metallic bond coat (MCrAlY; M = Co, Ni or Co/Ni), a ceramic top coat (YSZ, yttria-stabilized zirconia), and a thermally grown oxide layer (TGO) that forms at the metallic bond coat-ceramic top coat interface.

Point 3: Line 49: Citing [17] is not relevant here. This paper is only about shot peening.

Response 3: According to your comment, related references added in the manuscript.

Karaoglanli, A.C., Turk, A., Isothermal oxidation behavior and kinetics of thermal barrier coatings produced by cold gas dynamic spray technique, Surface and Coatings Technology, 318, 72-81, 2017.

Karaoglanli, A.C., Grund, T., Turk, A., Lampke, T., A comparative study of oxidation kinetics and thermal cyclic performance of thermal barrier coatings (TBCs), Surface and Coatings Technology, 371, 47-67, 2019.

Point 4: Line 55-56: What does it mean: “…Used for faster, easier, denser and more economical coating structure producing techniques are separated each other’s microstructural and  mechanical way [20]”? English is not proper.

Response 4: According to your comment, the sentence was changed and re-written in the manuscript.

The SSPS process enables the production of coatings cheaper, faster and easier. The D-gun technique also enables the production of fast and denser coating structure.

Point 5: Line 65-66: What does it mean “thinner microstructure“ in the sentence “…When compared to traditional plasma spray yttria stabilized zirconia (YSZ) coatings, SSPS coating has a thinner microstructure and higher  adhesion strength”? The sentence is not completed. Looks like you need to add: “…. as compared to that of….”

Response 5: According to your comment, the sentence was changed and re-written in the manuscript.

SSPS technique enables the production of more dense and high bonding strength coatings due to the shorter exposure of atmospheric conditions and faster production compared to the conventional plasma spray coating process.

Point 6: Line 75-77: Unfortunately authors do not clearly set up the paper tasks. “High temperature wear resistance” is defined as a function of “coating production technique”, “microstructural properties” and “wear mechanisms due to temperature and load”. Please, take a look at the sentence: “It has been observed that the high temperature wear resistance of the coatings varies according to the  coating production technique, the microstructural properties of the coating, as well as the wear  mechanisms due to temperature and load”. It is completely not correct. What does it mean: “microstructural properties” ?

Response 6: According to your comment, the sentence was changed and re-written in the manuscript.

As a result of the studies, it has been understood that the high-temperature wear performances of the coatings vary depending on the technique used in the production of coatings and microstructural characteristics that are generated due to the production process.

Point 7: Line 92: Please, put the scale on the images.Figure 1.Macro image of CoNiCrAlY coated substrates after wear tests: (a) produced by SSPS technique; (b) produced by D-gun technique.

Response 7: According to your comment, the scale in Figure 1 added in the manuscript.

Point 8: Line Lines1132-133: Unfortunately it is very difficult to understand: “…It can be seen from the microstructures that the CoNiCrAlY coating structure produced by the D-gun technique has a much more oxidized structure and the internal porosity content is lower than the CoNiCrAlY coating structure produced by the SSPS coating technique”. It means that “…internal porosity content is lower than the CoNiCrAlY coating structure”. How to compare porosity and structure????

Response 8: According to your comment, the sentence was changed and added in the manuscript.

The as-sprayed SSPS-CoNiCrAlY coatings exhibited a porous microstructure having micro-cracks as expected from a typical plasma spraying microstructure. As expected from the CoNiCrAlY metallic bond layer coated with the D-gun technique, the typical properties such as high hardness and low porosity were obtained from SEM images.

Point 9: Lines 134-138: The similar sentence: “It is seen that the surface roughness of the coating structure produced by the SSPS coating technique has a lower roughness value compared to the process produced by the D-gun coating technique. A higher level of oxide formation is the primary factor affecting the oxide formation on interface depending on coating production technique, temperature and time of experimental procedures [3,24,25]. It causes damage formation that result in lower level of oxidation resistance and detaching of the coating from the surface [31]”. The unclear terms are followings: “surface roughness of the coating structure”, “…It is seen that the surface roughness…. has a lower roughness”, “lower level of oxidation resistance”, “level… of detaching of the coating from the surface”.

Response 9: According to your comment, the sentence was changed and added in the manuscript.

It is seen that the surface roughness of the coating structure produced by the SSPS coating technique has a lower roughness value compared to the process produced by the D-gun coating technique from SEM images. This situation occurs depending on the production characteristics of the coatings and the production conditions.

Point 10: Line 296-297 How to understand the sentence in conclusion “…As a result of the analysis, it was observed that when the coatings produced by SSPS technique were tested at rt temperature was separated from the surface owing to regional breaks and spalling, this structure formed after cleaning the particles on the surface and forming a compact structure”?

Response 10: According to your comment, the conclusion part was changed and re-writed in the manuscript.

In this study, metallic bond coatings with CoNiCrAlY content were produced on Inconel 718 superalloy substrate material by using D-gun and SSPS coating techniques. The coatings produced were subjected to wear tests at three different temperatures: rt, 250°C and 500°C and different loads 2N and 5N in order to understand the wear mechanisms. The salient conclusions arising from this work are as follows:

  1. CoNiCrAlY metallic coating powders were successfully deposited on the nickel-based Inconel 718 superalloy substrate using D-Gun and SSPS coating techniques.
  2. The high-temperature wear behavior of the coatings has changed depending on the processes used in the coating production and the microstructural properties of the coatings after production.
  3. Depending on the increasing loading rates and temperature, wear losses were likewise increased. However, this increase was not linear.
  4. It has been understood that at 250 °C and rt surface fatigue-wear by using D-gun technique are comparatively more severe than SSPS technique.
  5. It has been observed that tribological layers and superficial changes occur in the microstructures of the coatings due to temperature and time by both thermal spray coating techniques.
  6. When high-temperature wear behaviors of CoNiCrAlY coatings are examined, it is seen that D-gun coatings show superior properties compared to SSPS coatings.
  7. Due to the increased surface hardness and microstructural dense structure with high-temperature effect, the wear resistance of the coatings increases.

Point 11: Lines 139-290. Unfortunately, the similar mistakes, unclear statements and terminology mistakes are in all manuscript chapters.

Response 11: According to your comment, the sentences were changed and added in the manuscript.

Round 2

Reviewer 2 Report

This version is ok.

Author Response

Response to Reviewer 3 Comments

Dear Reviewer,

The authors thank Reviewer 3 to his/her valuable contributions. We are also grateful for the positive evaluations of Reviewer 3.

Point 1: Line 24-25: The sentence “…The results show that the high temperature wear resistance of the coatings has changed with the microstructural and surface features of the samples depending on the coating production process and wear mechanisms” is not completely clear. About what surface features are you talking? Please, specify the new specific features of studied wear mechanisms in Abstract. English is not proper.

Authors write: “The high-temperature wear behaviour of the coatings has changed depending on the processes used in the coating production and the microstructural properties of the coatings after production.” The term “microstructural properties” is not correct

Response 1: According to your comment, related abstract part re-written and added in the manuscript.

The high-temperature wear behavior of the coatings has changed depending on the processes used in the coating production.

Point 2: Line 45-47: The sentence “…TBC systems are generally consisted from chemically stabile superalloy base material, CoNiCrAIY content owner high oxidation and corrosion resistance metallic bonding coating, porosity content owner heat insulating ceramic top coating and thermally growing oxide layer (TGO) consisted in bond and top coating interface during operating conditions” is not correct and clear. What does it mean “…CoNiCrAIY content owner, …porosity content owner ”? “TBC systems” do not consist of superalloy base material. It consists of Ceramics (ZrO2) and CoNiCrAIY bond layers.

Authors write: “Generally, a TBC system consists of a metallic bond coat (MCrAlY; M = Co, Ni or Co/Ni), a ceramic top-coat (YSZ, yttria-stabilized zirconia), and a thermally grown oxide layer (TGO) that forms at the metallic bond coat-ceramic top coat interface [9,18].” English is not understandable. What does it mean “…that forms at the metallic bond coat-ceramic top coat interface”?

Response 2: According to your comment, related introduction part re-written and added in the manuscript.

Generally, a TBC system consists of a metallic bond coat (MCrAlY; M = Co, Ni or Co/Ni), a ceramic top-coat (YSZ, yttria-stabilized zirconia), and a thermally grown oxide layer (TGO). The ceramic top-coat has an essentially low thermal conductivity and low thermal expansion and the metallic bond coat is deposited between the metallic substrate and ceramic top-coat for developing the adherence of the ceramic top coating to the substrate alloy [9,16,18].

Point 3: Line 49: Citing [17] is not relevant here. This paper is only about shot peening.

Response 3: According to your comment, related reference added in the manuscript.

Point 4: Line 50-56: What does it mean: “…Used for faster, easier, denser and more economical coating structure producing techniques are separated each other’s microstructural and mechanical way [20]”? English is not proper.

New authors edition is : “Improving the high-performance wear-resistant coating materials to protect the metallic substrate is an efficient approach to reducing wear. For this purpose, the wear rate and friction coefficient are significant parameters in of the performance of the coated components subjected to contact with others at high temperature. Besides, wear performances are generally related to the surface hardness of the material. For instance, wear resistance is especially important at blade tips, which run into a worn layer on the inner turbine shroud for perfect sealing. For that reason, the MCrAlY coatings have been used as the protective APS coatings against frictional wear at higher temperatures [20-22].” This general text does not relate MCrAlY coatings. There is no any logic in these sentences. What does it mean ”… For this purpose, the wear rate and friction coefficient are significant parameters…”? These parameters are well known for all sliding processes. Authors did not prove any issues related to MCrAlY coatings

Response 4: The indicated sentence has been changed according to your valuable comment.

Improving the high-performance wear-resistant coating materials to protect the metallic substrate is an efficient approach to reducing wear. MCrAlY overlay coating is used as a protective coating against high-temperature oxidation and corrosion. Due to the sufficient amount of Al in the coatings during service life is the main degradation factor in conventional MCrAlY coatings. Performance of a metallic bond coat can be improved by applying Al and Cr gradient MCrAlY coatings [20].

Point 5: Line 65-66: What does it mean “thinner microstructure“ in the sentence “…When compared to traditional plasma spray yttria stabilized zirconia (YSZ) coatings, SSPS coating has a thinner microstructure and higher adhesion strength”? The sentence is not completed. Looks like you need to add: “…. as compared to that of….”

Response 5: According to your comment, the sentence was changed and re-written in the manuscript.

SSPS technique enables the production of more dense and high bonding strength coatings due to the shorter exposure of atmospheric conditions and faster production compared to the conventional plasma spray coating process.

Point 6: Line 75-77: Unfortunately authors do not clearly set up the paper tasks. “High temperature wear resistance” is defined as a function of “coating production technique”, “microstructural properties” and “wear mechanisms due to temperature and load”. Please, take a look at the sentence: “It has been observed that the high temperature wear resistance of the coatings varies according to the coating production technique, the microstructural properties of the coating, as well as the wear mechanisms due to temperature and load”. It is completely not correct. What does it mean: “microstructural properties” ?

Unfortunately, authors did not answer for above questions and did not correct mistakes underlined below in the author’s edition: “Thermal spray coatings have been widely utilized in various industrial applications against surface damages such as wear corrosion and oxidation. Therefore, low porosity and good adhesion behavior are desired for the coating. This work aims to investigate the microstructural mechanical properties and abrasion behaviors of CoNiCrAlY-based metallic bond coats onto nickel-based superalloy substrate Inconel 718 using the D-gun and SSPS coating techniques. To analyse the high-temperature wear behavior of the coatings, wear tests were applied at different temperatures and different loads. Before and after high-temperature wear tests microstructural properties, mechanical properties and surface properties of CoNiCrAlY coatings were examined. SEM, EDS, hardness and 3D-profilometerof coated and abraded surfaces were examined comparatively. As a result of the studies, it has been understood that the high-temperature wear performances of the coatings vary depending on the technique used in the production of coatings and microstructural characteristics that are generated due to the production process.”

Response 6: According to your comment, the sentence was changed and re-written in the manuscript.

Point 7: Line 92: Please, put the scale on the images. Figure 1.Macro image of CoNiCrAlY coated substrates after wear tests: (a) produced by SSPS technique; (b) produced by D-gun technique.

Response 7: According to your comment, the scale in Figure 1 added in the manuscript.

Point 8: Line 110-111. There is no description of roughness, hardness and porosity determination

Response 8: According to your comment, description of roughness, hardness and porosity explanation are added in the manuscript.

Using the contact stylus instrument Hommelwerke device, the average surface roughness values were obtained by measuring from 5 to 10 different points on each sample according to the standard. Microhardness measurements of litter materials and coatings were made in Duramin brand test device according to DIN EN ISO 4516 norm. Porosity measurements of the coatings were made by defining the matrix and porosity structures in microstructures in the image analysis program.

Point 9: Lines132-133: Unfortunately it is very difficult to understand: “…It can be seen from the microstructures that the CoNiCrAlY coating structure produced by the D-gun technique has a much more oxidized structure and the internal porosity content is lower than the CoNiCrAlY coating structure produced by the SSPS coating technique”. It means that “…internal porosity content is lower than the CoNiCrAlY coating structure”. How to compare porosity and structure????

Response 9: According to your comment, the sentence was changed and added in the manuscript.

The as-sprayed SSPS-CoNiCrAlY coatings exhibited a porous microstructure having micro-cracks as expected from a typical plasma spraying microstructure. As expected from the CoNiCrAlY metallic bond layer coated with the D-gun technique, the typical properties such as high hardness and low porosity were obtained from SEM images.

Point 10: Lines140-169: Authors suggest the following edition which has mistakes and unclear issues shown by bold and underlined fonts: "...Figure 3a shows the microstructure of the coating deposited by the D-gun technique. The porosities, oxide formations and surface roughness differences of the coatings produced by SSPS and D-Gun techniques can be seen in SEM microstructures. The as-sprayed SSPS-CoNiCrAlY coatings exhibited a porous microstructure having micro-cracks as expected from a typical plasma spraying microstructure. As expected from the CoNiCrAlY metallic bond layer coated with the D-gun technique, the typical properties such as high hardness and low porosity were obtained from SEM images. It is seen that the surface roughness of the coating structure produced by the SSPS coating technique has alower roughness value compared to the process produced by the D-gun coating technique from SEM images. This situation occurs depending on the production characteristics of the coatings and the production conditions [3,28,29]. It causes damage formation that results in a lower level of oxidation resistance and detaching of the coating from the surface {35]. The hardness, surface roughness, porosity and oxide content values of CoNiCrAlY metallic bond coatings produced by D-gun and SSPS coating techniques are specified in Table 3. From Figure 3, it is seen that CoNiCrAlY metallic bond coatings produced by using the D-gun coating technique are denser than metallic bond coatings produced by SSPS coating technique. After the dry sliding wear tests performed at different temperatures and different loads, the volume loss graphs obtained with the 3D-profilometer are given in Figure 4. Generally, the samples produced by the D-gun technique showed better wear resistance both at rt and at 250 °C and 500 °C [7]. It is possible to associate the first reason for this with hardness. The other effective parameter is microstructural features that come from the production process and conditions. Hardness and strength values of coatings produced by using the D-gun technique are comparatively higher than other thermal spray coating techniques [36]. When the coatings produced by D-gun and SSPS techniques were evaluated microstructurally, it is seen from the microstructures that the coatings produced by the SSPS technique have higher porosity content and surface roughness. In the D-gun coating technique, denser coating structure is obtained because of the high speed used in the production of coatings [34]. The microstructural defects, high porosity structure of SSPS coatings have had a negative effect on the abrasion, strength and toughness properties of the coatings. The techniques used in the production of coatings are mainly effective in their performance during usage conditions [37]."

The main questions are followings: How to expect properties from the from the CoNiCrAlY metallic bond layer? How to obtain high hardness and low porosity from SEM images??? How to compare lower roughness value with the process??? It is impossible to see from Fig.3 that that “…CoNiCrAlY metallic bond coatings produced by using the D-gun coating technique are denser than metallic bond coatings produced by SSPS coating technique”because of low resolution and magnification of the images and absence of the measurement data. It is very difficult to find the porosity difference of 0.3% by any experimental methods (Table 3) when the values are 1.2 ±1.0. So, the discussion about differences of coating density looks not true.

Response 10: According to your comment, the sentence was changed and all explanations are added in the manuscript.

Reviewer 3 Report

Review of Corrected Manuscript (08.07.2020)

Microabrasion, dry sliding and high temperature  wear resistance of MCrAlY coatings deposited by detonation gun (D-gun) and supersonic plasma  spraying (SSPS) techniques

Unfortunately, in many cases authors only shortened sentences without additional explanations and clarifications. Some mistakes are same (for example “microstructural  properties” )  The previous and new comments are below.

  1. Line 24-25: The sentence “…The results show that the high  temperature wear resistance of the coatings has changed with the microstructural and surface  features of the samples depending on the coating production process and wear mechanisms” is not completely clear. About what surface features are you talking?  Please, specify  the new specific features of studied wear mechanisms in Abstract. English is not proper.                                                                            Authors write: “The high-temperature wear behaviour of the coatings  has changed depending on the processes used in the coating production and the microstructural  properties of the coatings after production.” The term “microstructural  properties” is not correct
  1. Line 45-47: The sentence “…TBC systems are generally consisted from chemically stabile  superalloy base material, CoNiCrAIY content owner high oxidation and corrosion resistance  metallic bonding coating, porosity content owner heat insulating ceramic top coating and thermally  growing oxide layer (TGO) consisted in bond and top coating interface during operating conditions” is not correct and clear. What does it mean “…CoNiCrAIY content owner, …porosity content owner ”? “TBC  systems” do not consist of superalloy base material. It consists of Ceramics (ZrO2) and CoNiCrAIY bond layers.

Authors write: “Generally, a  TBC system consists of a metallic bond coat (MCrAlY; M = Co, Ni or Co/Ni), a ceramic top-coat (YSZ,  yttria-stabilized zirconia), and a thermally grown oxide layer (TGO) that forms at the metallic bond coat-ceramic top coat interface [9,18].” English is not understandable. What does it mean “…that forms at the metallic bond coat-ceramic top coat interface”?

  1. Line 49: Citing [17] is not relevant here. This paper is only  about shot peening.
  1. Line 50-56: What does it mean: “…Used for faster, easier, denser and more economical coating structure producing techniques are separated each other’s microstructural and  mechanical way [20]”?  English is not proper.

New authors edition is : “Improving the high-performance wear-resistant coating materials to protect the metallic substrate is an efficient approach to reducing wear. For this purpose, the wear rate and friction coefficient are significant parameters in of the performance of the coated components subjected to contact with others at high temperature. Besides, wear performances are generally related to the surface hardness of the material. For instance, wear resistance is especially important at blade tips, which run into a worn layer on the inner turbine shroud for perfect sealing. For that reason, the  MCrAlY coatings have been used as the protective APS coatings against frictional wear at higher  temperatures [20-22].” This general text does not  relate MCrAlY coatings. There is no any logic in these sentences. What does it mean ”… For this purpose, the wear rate and friction coefficient are significant parameters…”?  These parameters are well known for all sliding processes.  Authors did not prove any issues related to MCrAlY coatings  

  1. Line 65-66: What does it mean “thinner microstructure“ in the sentence “…When compared to traditional plasma spray yttria stabilized zirconia (YSZ) coatings, SSPS coating has a thinner microstructure and higher  adhesion strength”? The sentence is not completed. Looks like you need to add: “…. as compared to that of….”  
  2. Line 75-77: Unfortunately authors do not clearly set up the paper tasks. “High temperature wear resistance” is defined as a function of “coating production technique”, “microstructural properties” and “wear  mechanisms due to temperature and load”. Please, take a look at the sentence: “It has  been observed that the high temperature wear resistance of the coatings varies according to the  coating production technique, the microstructural properties of the coating, as well as the wear  mechanisms due to temperature and load”. It is completely not correct. What does it mean: “microstructural properties” ?

Unfortunately, authors did not answer for above questions and did not correct mistakes underlined below in the author’s edition: “Thermal spray coatings have been widely utilized in various industrial applications against  surface damages such as wear corrosion and oxidation. Therefore, low porosity and good adhesion  behavior are desired for the coating. This work aims to investigate the microstructural mechanical  properties and abrasion behaviors of CoNiCrAlY-based metallic bond coats onto nickel-based  superalloy substrate Inconel 718 using the D-gun and SSPS coating techniques. To analyse the  high-temperature wear behavior of the coatings, wear tests were applied at different temperatures  and different loads. Before and after high-temperature wear tests microstructural properties,  mechanical properties and surface properties of CoNiCrAlY coatings were examined. SEM, EDS,  hardness and 3D-profilometer of coated and abraded surfaces were examined comparatively. As a  result of the studies, it has been understood that the high-temperature wear performances of the  coatings vary depending on the technique used in the production of coatings and microstructural  characteristics that are generated due to the production process.”

  1. Line 92: Please, put the scale on the images. Figure 1. Macro image of CoNiCrAlY coated substrates after wear tests: (a) produced by SSPS  technique; (b) produced by D-gun technique.  
  2. Line 110-111. There is no description of roughness, hardness and porosity determination
  3. Lines132-133: Unfortunately it is very difficult to understand: “…It can be seen from the microstructures that the CoNiCrAlY coating structure produced by the D-gun technique has a much more oxidized  structure and the internal porosity content is lower than the CoNiCrAlY coating structure produced  by the SSPS coating technique”. It means that “…internal porosity content is lower than the CoNiCrAlY coating structure”. How to compare porosity and structure????
  4. Lines140-169: Authors suggest the following edition which has mistakes and unclear issues shown by bold and underlined fonts: "...Figure 3a shows the microstructure of the coating deposited by the D-gun technique. The porosities, oxide formations and surface roughness differences of the coatings produced by SSPS  and D-Gun techniques can be seen in SEM microstructures. The as-sprayed SSPS-CoNiCrAlY  coatings exhibited a porous microstructure having micro-cracks as expected from a typical plasma  spraying microstructure. As expected from the CoNiCrAlY metallic bond layer coated with the  D-gun technique, the typical properties such as high hardness and low porosity were obtained from SEM images. It is seen that the surface roughness of the coating structure produced by the SSPS  coating technique has a lower roughness value compared to the process produced by the D-gun  coating technique from SEM images. This situation occurs depending on the production characteristics of the coatings and the production conditions [3,28,29]. It causes damage formation  that results in a lower level of oxidation resistance and  detaching of the coating from the surface {35]. The hardness, surface roughness, porosity and oxide content values of CoNiCrAlY metallic  bond coatings produced by D-gun and SSPS coating techniques are specified in Table 3. From Figure  3, it is seen that CoNiCrAlY metallic bond coatings produced by using the D-gun coating technique  are denser than metallic bond coatings produced by SSPS coating technique. After the dry sliding wear tests performed at different temperatures and different loads, the volume loss graphs obtained with the 3D-profilometer are given in Figure 4. Generally, the samples produced by the D-gun technique showed better wear resistance both at rt and at 250 °C and 500 °C  [7]. It is possible to associate the first reason for this with hardness. The other effective parameter is  microstructural features that come from the production process and conditions. Hardness and  strength values of coatings produced by using the D-gun technique are comparatively higher than  other thermal spray coating techniques [36]. When the coatings produced by D-gun and SSPS  techniques were evaluated microstructurally, it is seen from the microstructures that the coatings  produced by the SSPS technique have higher porosity content and surface roughness. In the D-gun coating technique, denser coating structure is obtained because of the high speed used in the  production of coatings [34]. The microstructural defects, high porosity structure of SSPS coatings  have had a negative effect on the abrasion, strength and toughness properties of the coatings. The  techniques used in the production of coatings are mainly effective in their performance during usage  conditions [37]."                          The main questions are followings: How to expect properties from the from the CoNiCrAlY metallic bond layer? How to obtain high hardness and low porosity from SEM images??? How to compare lower roughness value with the process??? It is impossible to see from Fig.3 that that “…CoNiCrAlY metallic bond coatings produced by using the D-gun coating technique  are denser than metallic bond coatings produced by SSPS coating technique”because of low resolution and magnification of the images and absence of the measurement data.  It is very difficult to find the porosity difference of 0.3% by any experimental methods (Table 3) when  the values are 1.2 ±1.0.  So, the discussion about differences of coating density looks not true. 

Author Response

(The authors gave the same response as above.)
